# Constraint Loss for Rotated Object Detection in Remote Sensing Images

Luyang Zhang [1] , Haitao Wang [1], Lingfeng Wang [2,3,*], Chunhong Pan [3], Qiang Liu [1] and Xinyao Wang [1]

1 College of Automation Engineering, Nanjing University of Aeronautics & Astronautics, Nanjing 210016, China; zhangly2020@nuaa.edu.cn (L.Z.); htwang@nuaa.edu.cn (H.W.); liuqiang21@nuaa.edu.cn (Q.L.); xinyaowang@nuaa.edu.cn (X.W.)
2 College of Information Science and Technology, Beijing University of Chemical Technology, Beijing 100029, China
3 National Laboratory of Pattern Recognition, Institute of Automation, Chinese Academy of Sciences, Beijing 100190, China; chpan@nlpr.ia.ac.cn
* Correspondence: lfwang@mail.buct.edu.cn; Tel.: +86-010-82544592

**Abstract:** Rotated object detection is an extension of object detection that uses an oriented bounding box instead of a general horizontal bounding box to define the object position. It is widely used in remote sensing images, scene text, and license plate recognition. The existing rotated object detection methods usually add an angle prediction channel in the bounding box prediction branch, and smooth $L_1$ loss is used as the regression loss function. However, we argue that smooth $L_1$ loss causes a sudden change in loss and slow convergence due to the angle solving mechanism of open CV (the angle between the horizontal line and the first side of the bounding box in the counter-clockwise direction is defined as the rotation angle), and this problem exists in most existing regression loss functions. To solve the above problems, we propose a decoupling modulation mechanism to overcome the problem of sudden changes in loss. On this basis, we also proposed a constraint mechanism, the purpose of which is to accelerate the convergence of the network and ensure optimization toward the ideal direction. In addition, the proposed decoupling modulation mechanism and constraint mechanism can be integrated into the popular regression loss function individually or together, which further improves the performance of the model and makes the model converge faster. The experimental results show that our method achieves 75.2% performance on the aerial image dataset DOTA (OBB task), and saves more than 30% of computing resources. The method also achieves a state-of-the-art performance in HRSC2016, and saved more than 40% of computing resources, which confirms the applicability of the approach.

**Keywords:** rotated object detection; remote sensing image; loss functions; fast convergence

## 1. Introduction

Remote sensing images are an important manifestation of remote sensing information, which is vital to national defense security. Remote sensing-image object detection is a prerequisite and basis for tasks such as spatial object tracking and instance segmentation [1]. With the extensive application of convolutional neural networks (CNNs) in computer vision, object detection has undergone rapid development [2]. R-CNN [3] is predominantly used for object detection based on deep learning. After 2016, a series of two-stage detectors based on candidate regions have become the mainstream, such as Fast RCNN [4], Faster RCNN [5], and R-FCN [6]. The two-stage detector has good detection accuracy; however, the detection speed is poor owing to the complex network structure. YOLO [7], SSD [8], and RetinaNet [9] are representative one-stage detectors; they do not involve the region proposal network and greatly improved the detection speed; however, the detection accuracy is sacrificed. While the anchor-based method is developing rapidly, anchor-free methods have also received attention in recent years owing to the proposal of CornerNet [10]. The more popular ones

are FCOS [11], CenterNet [12], and ExtremeNet [13]. They have replaced the previous generation of anchor methods by predicting key points, thereby opening up a new direction for the research of object detection technology [14]. In addition, there are some researches on high resolution [15,16], unbalanced samples [17], and other issues [18]. The above method has achieved good performance in natural images, such as the COCO [19] and Pascal VOC [20] datasets. Therefore, it is applied to remote sensing image object detection tasks. For example, Zhang et al. [21] combined with fast registration and YOLOv3, proposed an effective aerial infrared image sequence moving vehicle detection method. Liao et al. [22] proposed the Local Perception Region Convolutional Neural Network (LR-CNN) and constructed a new method for vehicle detection in aerial images. Lei et al. [23] proposed a tiny vehicle detection method based on spatio-temporal information, which realized the detection of tiny moving vehicle in satellite video. However, the horizontal bounding box cannot provide accurate orientation and scale information in remote-sensing-image object detection tasks [24–26] (see Figure 1). Therefore, the research of rotated object detection in remote sensing images is of great significance for engineering applications.

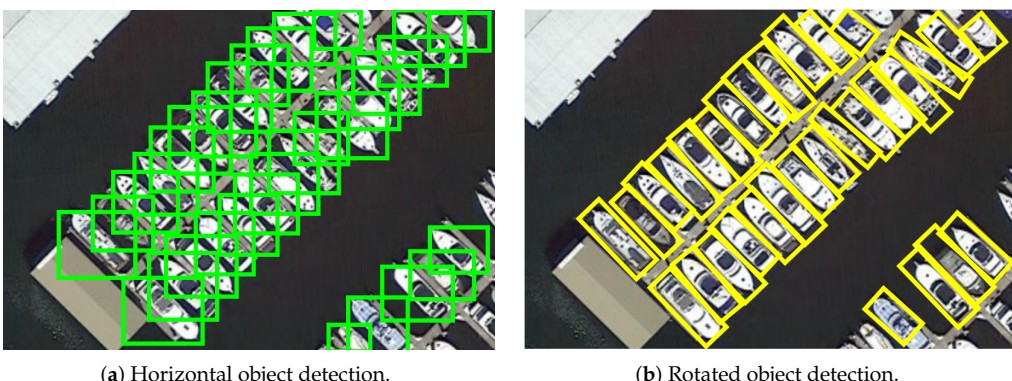

(**a**) Horizontal object detection.  (**b**) Rotated object detection.

**Figure 1.** Comparison results of horizontal object detection and rotating object detection. Rotated object detection plays an important role in remote-sensing-image object detection.

In recent years, rotated object detection has been derived from classic object detection [27–29], and most existing methods use five parameters (coordinates of the central point, width, height, and rotation angle) to describe the oriented bounding box. The initial exploration of rotated object detection involves rotating the RPN [30]; however, it involves more anchors, which implies that additional running time is required. Ding et al. [31] proposed an RoI transformer that converts the axis-aligned RoI into a rotatable RoI to solve the problem of misalignment between the RoI and the oriented object. Han et al. [32] proposed a $S^2$ANet, which was used for depth feature alignment for rotating object detection. In addition, SCRDet [33] reported for the first time the problem of sudden changes in loss in rotated object detection tasks (see Figure 2) and proposed IoU-smooth $L_1$ loss to overcome this problem. Similarly, PIoU Loss [34] and $R^3$Det [35] both add a very small weight to the loss function to overcome the problem of sudden change in loss. However, these methods all inhibit the sudden change in loss, and do not solve the problem fundamentally. Therefore, some novel ideas have been proposed. Zhu [26] and Xu [36], respectively, proposed a new method for expressing directed objects in aerial images, which avoids complicated calculation rules, but the performance is not ideal. In CSL [37], a circular label is designed to convert the angle-regression problem into a classification problem. In RSDet [38], a modulation rotation loss is proposed to eliminate the problem of discontinuity in loss. Although CSL and RSDet both effectively solve the problem of loss mutation, there are still some new problems that have not been considered. For example, the detection performance of CSL is not ideal; RSDet does not consider the problem of training resource consumption. Therefore, a new regression loss mechanism is essential for the development of rotating object detection.

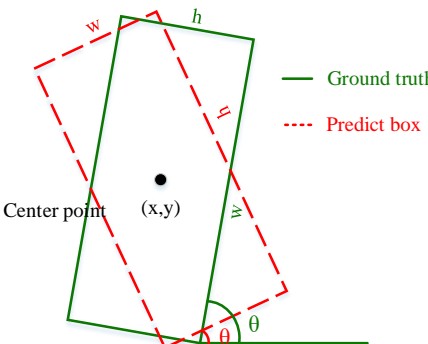

**Figure 2.** Reasons for sudden changes in loss. Only the angle $\theta$ regression is considered, assuming that the center point and size of the prediction frame and ground truth are the same, and size of the long side and the short side are 30 and 15, respectively. Consequently,the above bounding box can be described by five parameters: the prediction box (x, y, 30, 15, 85°) and the ground truth box (x, y, 15, 30, 5°). The prediction offset is: (0, 0, 15, 15, 80°) and the ideal offset is (0, 0, 0, 0, 10°). It can be seen that $L_1$ loss is far greater than ideal owing to the exchange of width and height and the periodicity of the angle.

To solve the above problems, we propose a new loss function with a decoupling modulation mechanism and constraint mechanism. The decoupling modulation mechanism divides the deviation of smooth $L_1$ loss into three parts (center point, size, and rotation angle) and modulates them, which effectively overcomes the sudden change in loss. On this basis, the constraint mechanism provides a constraint domain for the center point and size of the bounding box so that it has a tolerance for deviation in the regression process. This improves the performance and convergence speed of the model. In addition, the decoupling modulation mechanism and constraint mechanism we proposed are general and they perform well when applied to most popular regression loss functions.

In summary, the contributions of this paper are as follows:

- We propose a decoupling modulation mechanism that decouples the loss deviation into three parts and modulates them, respectively. It overcomes the problem of sudden changes in loss for detecting rotating objects and makes the training process more stable.
- We propose a constraint mechanism, which effectively solves the problem of slow network convergence and improves the performance of the model by adding the constraint domain as the center point and size of the bounding box. Experiments on the DOTA dataset reveal an improvement in mAP by 1.2% and in convergence speed by 40%, and the 0.5% mAP and 30% convergence speed are improved on the HRSC2016 dataset.
- Our method is independent of the model; thus, it is generic and can be applied to most regression loss functions. The experimental results for nine popular loss functions (including deviation-based loss and IoU-based loss) verify its effectiveness.

The remainder of this paper is organized as follows. Section 2 describes our motivation, the proposed method, and a detailed analysis of the characteristics of the method. Section 3 reports the details of the experiment, including the datasets, implementation details, ablation study, and experimental results. Finally, Section 4 presents the conclusions of this article.

## 2. Materials and Methods

In this section, we first describe the proposed constraint loss function and then analyze the constraint parameters (CPs). Finally, the adjustability and generalizability of the constraint loss are discussed.

### 2.1. Constrained Loss Function

In the initial stage of the development of rotating object detection, smooth $L_1$ loss still plays an important role and the regression is represented in (1).

$$
\begin{aligned}
t_x &= (x - x^*)/w, t_y = (y - y^*)/h \\
t_w &= log(w^*/w), t_h = log(h^*/h) \\
t_\theta &= \theta - \theta^*
\end{aligned}
\tag{1}
$$

where $(x, y)$ is the coordinate of the center point of the rectangular box, $(w, h)$ represents its width and height, and $\theta$ is defined as the acute angle to the X-axis. The range of values of $\theta$ is $[0, \frac{\pi}{2}]$ or $[-\frac{\pi}{2}, \frac{\pi}{2}]$, as defined by openCV (see Figure 3). The $^*$ represents the ground truth labels, and the regression calculation is normalized to avoid overfitting.

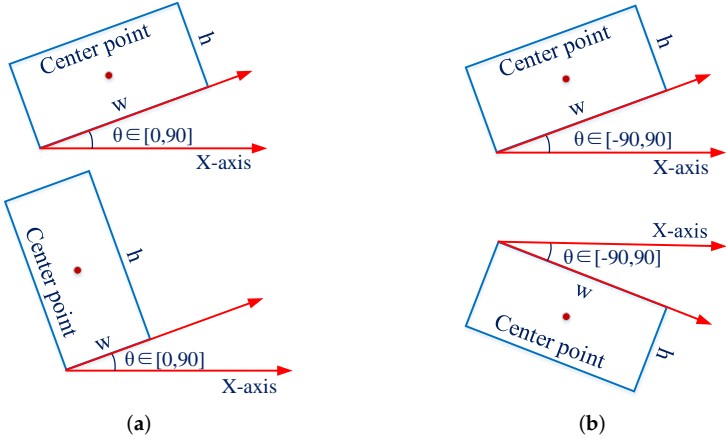

(a)  (b)

**Figure 3.** Bounding box five-parameter regression method: (**a**) $90°$ regression method; (**b**) $180°$ regression method.

The sudden change in loss of the oriented bounding box regression process is mainly caused by two reasons: (i) the exchange of width and height, and (ii) the periodicity of the angle. To solve the above two problems, we decoupled and modulated the regression calculation of the rotating bounding box.

Decoupling and Modulation: Inspired by [38], the regression calculation of the oriented bounding box is first decoupled into three parts: (i) center point regression, (ii) size regression, and (iii) angle regression. Then, the latter two parts are modulated as follows:

$$
L_1 = |x - x^*| + |y - y^*|
\tag{2}
$$

$$
L_2 = \min \begin{cases} |w - w^*| + |h - h^*| \\ |w - h^*| + |h - w^*| \end{cases}
\tag{3}
$$

$$
L_3^{90} = \min \begin{cases} |\theta - \theta^*| \\ \dfrac{\pi}{2} - |\theta - \theta^*| \end{cases}
\tag{4}
$$

In [38], the exchange of edges is always accompanied by modulation in the angle period, and the bounding box regression is expressed as:

$$
L_{reg} = \min \begin{cases} L_1 + L_2^1 + L_3^1 \\ L_1 + L_2^2 + L_3^2 \end{cases}
\tag{5}
$$

where $L_2^1$ represents the first row of $L_2$ in (3) (the same definitions for $L_2^2$, $L_3^1$, and $L_3^2$).

The above design is useful for the regression of most bounding boxes; however, some special problems are ignored. For example, the $L_2$ and $L_3$ modulations are not synchronized,

as shown in Figure 4. Therefore, the regression of the bounding box is divided into three parts, as shown in (2)–(4), and modulated, respectively. Finally, the output result is modulated twice, and the bounding box is mirrored and rotated. In addition, when the 180° representation method is used, $L_2$ modulation is suppressed, and the modulation term $\pi/2$ of $L_3$ is replaced by $\pi$.

$$L_3^{180} = \min \begin{cases} |\theta - \theta^*| \\ \pi - |\theta - \theta^*| \end{cases} \tag{6}$$

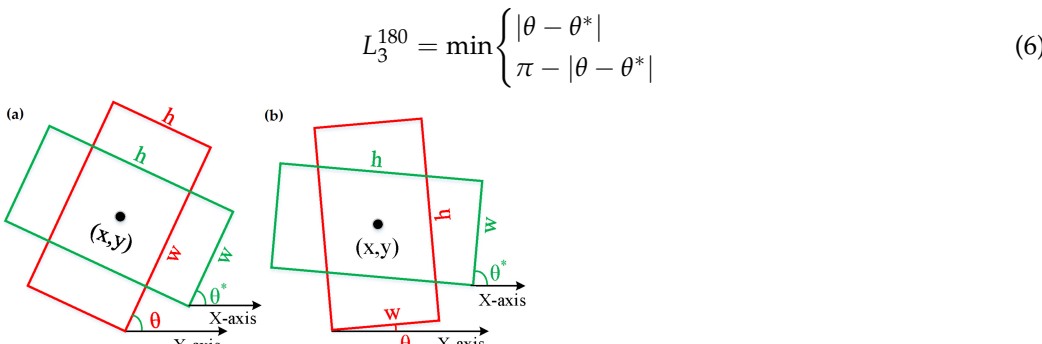

**Figure 4.** $L_2$ and $L_3$ modulations were not synchronized. (**a**) $L_2$ modulation and $L_3$ non-modulation. (**b**) $L_2$ non-modulation and $L_3$ modulation. Obviously, when 2 and 3 are modulated or not modulated at the same time, it is difficult for the prediction box to approximate the true value box. However, when the two are not synchronized, they can be quickly approached.

Constraint: To enable the network to converge faster and reduce resource consumption, the calculation of the deviation between the center point and the size of the bounding box must be constrained. The constraint is expressed as follows:

$$L_1^* = \text{con}(\sqrt{(x - x^*)^2 + (y - y^*)^2} > \alpha|L_1, 0) \tag{7}$$

$$L_2^* = \text{con}(\min \begin{cases} \sqrt{(w - w^*)^2 + (h - h^*)^2} \\ \sqrt{(w - h^*)^2 + (h - w^*)^2} \end{cases} > \beta|L_2, 0)$$

where, $\text{con}(.)$ is the conditional function, which means $L_i^* = L_i$ when the condition is met; otherwise, $L_i = 0$. $\alpha$ and $\beta$ represent the constraints on the center point and target frame scale, respectively, to ensure that the model always evolves in the correct direction during the training process.

The regression loss function is shown in Figure 5, where the constraint line comes from the Constraint Parameters (CPs) and coincides with the X-axis. In particular, it is abandoned below the constraint line. This means that the loss value is set to 0 when the deviation between the predicted box and the ground truth is within the constraint range.

In summary, the proposed constraint loss functions $L_C$ are expressed as follows:

$$
\begin{aligned}
L_C^{90} = {}& \text{con}(\sqrt{(x - x^*)^2 + (y - y^*)^2} > \alpha|L_1, 0) \\
+ {}& \text{con}(\min \begin{cases} \sqrt{(w - w^*)^2 + (h - h^*)^2} \\ \sqrt{(w - h^*)^2 + (h - w^*)^2} \end{cases} > \beta|L_2, 0) \\
+ {}& \min(|\theta - \theta^*|, \frac{\pi}{2} - |\theta - \theta^*|) \\
\end{aligned} \tag{8}
$$

$$
\begin{aligned}
L_C^{180} = {}& \text{con}(\sqrt{(x - x^*)^2 + (y - y^*)^2} > \alpha|L_1, 0) \\
+ {}& \text{con}(\sqrt{(w - w^*)^2 + (h - h^*)^2} > \beta|L_2, 0) \\
+ {}& \min(|\theta - \theta^*|, \pi - |\theta - \theta^*|) \\
\end{aligned} \tag{9}
$$

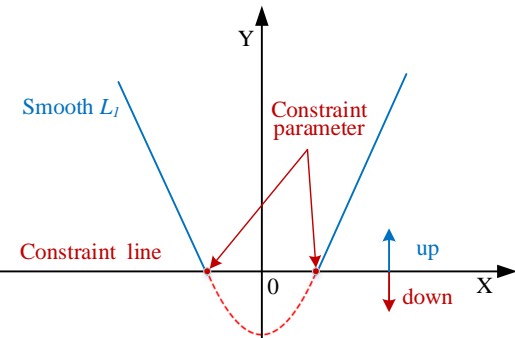

**Figure 5.** Schematic diagram of the loss function. The position of the restraint line is adjustable and determined by the CP.

### 2.2. Constraint Analysis

Without loss of generality, some changes occurred in the fine-tuning of the network owing to the introduction of $\alpha$ and $\beta$. We explored its geometric meaning and analyzed it using mathematical reasoning.

Geometric meaning: The geometric meaning of the CPs, proposed in this study, is shown in Figure 6. In the figure, $\alpha$ represents the constraint radius of the center point, and the center point deviation is set to 0 when the distance between the center point of the prediction box and the ground truth is less than $\alpha$. $\beta$ represents the bounding box size constraint radius, and the size deviation is set to 0 when the difference between the predicted box size and the ground truth is less than $\beta$. In addition, the endpoints of the bounding box are limited to the same color area. It is worth noting that this is similar to the filter of the bounding box, but it is completely different. Our goal is to prevent the prediction box from developing in a bad direction when the center point or size of the prediction box is close to the ground truth.

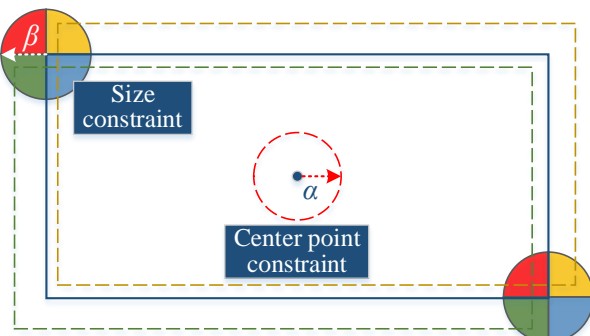

**Figure 6.** The geometric meaning of CPs. $\alpha$ is the constraint radius of center point; $\beta$ is the constraint radius of size. The endpoints of the bounding box are limited to the same color area.

Gradient Analysis: Figure 7 shows a simplified diagram of the network structure. In the process of loss back propagation, the convolutional layer, pooling layer, and fully connected layer occupy the dominant position. Therefore, we analyzed the gradients of the three key layers mentioned, and studied the influence of the constraint loss function on them.

The gradient of the loss function $S(L)$ with respect to the output layer is expressed as:

$$\delta^Y = \triangledown_L \odot \sigma'(z^Y) = L \odot \sigma'(z^Y) \tag{10}$$

where $\triangledown$ represents the gradient vector of the loss function S with respect to the predicted value Y, $\sigma'$ represents the partial derivative of the activation function $z(X)$, $\odot$ is the *Hadamard* product, which represents the point-to-point multiplication operation between matrices or vectors.

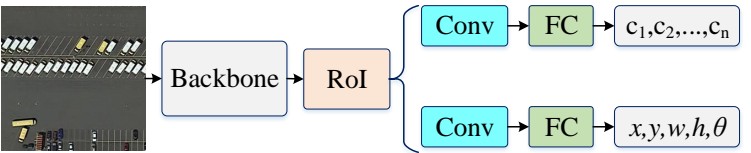

**Figure 7.** Image information transmission diagram. It mainly includes backbone network, RoI module, and head module.

The gradient propagation of the fully connected layer is expressed as:

$$\delta^l = ((W^{l+1})^T \delta^{l+1}) \odot \sigma'(z^l) \tag{11}$$

The input $P^{l-1}$ of the pooling layer can be obtained from $P^l$. This process is usually called an upsample.

$$P^{l-1} = upsample(P^l \odot \sigma'(z^{l-1})) \tag{12}$$

where the second term can be understood as the constant 1 in the pooling process, because no activation function is involved in the pooling layer.

The input $C^{l-1}$ of the convolutional layer can be obtained from $C^l$.

$$C^{l-1} = C^l * rot180(W^l) \odot \sigma'(z^{l-1}) \tag{13}$$

A more detailed gradient analysis is shown in Appendix A, and based on this, we know the influence of the constraint loss function on the key layer. When the center point constraint ($L_1^*$) or the size constraint ($L_2^*$) is activated, the components of the regression deviation $L_C$ are reduced, and this response is directly transferred to the calculation of the gradient value of the main layer ($\delta^Y$, $\delta^l$, $P^{l-1}$, $C^{l-1}$), and parameter adjustment without additional steps. This simplifies the backpropagation task. In particular, it is more pronounced when $L_1^*$ and $L_2^*$ are activated simultaneously. This means that only the angle parameter $\theta$ is adjusted ($L = L_3$), and the task will be simple and easy to implement. It is worth noting that when $L_1^*$ and $L_2^*$ are activated, the center point and size of the candidate frame are adjusted to the constraint range (see Figure 6), and subsequent adjustments have also been suppressed. This avoids additional calculations and development in unfavorable directions.

Convergence Analysis: Through geometric and gradient analyses, we found that the introduction of CPs had a positive effect on the convergence speed of the network. This is reasonable because the introduction of CPs is equivalent to increasing the tolerance of the prediction box. In the process of returning the prediction box to the truth box, the existence of tolerance enables the network to stabilize faster. To verify this idea, we tested the number of iterations when the network reached stability under different CPs, and the experimental results verified our hypotheses.

### 2.3. Adjustability and Generalizability

Adjustability: Inspired by self-placed learning [39], a cascaded sequence of CPs was proposed [40]. At the beginning of the training phase, a larger constraint parameter can filter out candidate frames with lower confidence, so that high-confidence bounding boxes can be focused on training. As the training progresses and the restriction parameters become smaller, the network begins to pay attention to the modification and optimization of the bounding box. In addition, the bounding box is restrained from expanding toward the constraint range, so that the network always develops in the ideal direction.

Generalizability: Based on the above analysis, we know that the introduction of $\alpha$ and $\beta$ can improve the performance of the regression loss function and prompt the network to always train in the right direction. Therefore, we consider applying this constraint method to other regression loss functions and analyze its generality.

The existing rotating bounding box regression loss function can be divided into two categories: deviation-based and IoU-based categories. The deviation-based loss functions include MSE, MAE, Huber [41], Log-Cosh [42], and Quantile [43]. IoU-based loss functions include IoU [44], GIoU [45], DIoU [46], and CIoU. For the deviation-based loss functions, a cascade constraint parameter sequence is set (see Figure 8). For the IoU-based loss functions, we designed an overlapping sequence as its CP sequence. When the IoU of the prediction box and ground truth is greater than the threshold, the loss is considered to be zero (general IoU loss = 1-IoU). The initial threshold of overlap is 0.5, and gradually increases to 0.75 as the number of training increases. A more detailed parameter design is presented in Table 1.

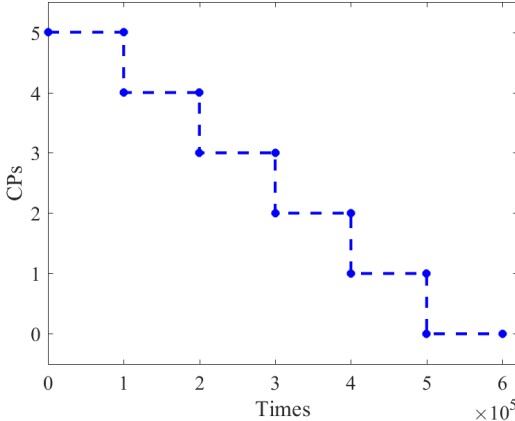

**Figure 8.** CPs. As the training progresses, the CPs $\alpha$ and $\beta$ decrease stepwise.

**Table 1.** The overlap threshold parameter sequence setting of the IoU-based loss functions.

| Stage ($\times 10^5$) | S0 | S1 | S2 | S3 | S4 | S5 |
|---|---|---|---|---|---|---|
| IoU threshold | 0.5 | 0.55 | 0.6 | 0.65 | 0.7 | 0.75 |

## 3. Experiments and Results

Our experiment was carried out on a server with Ubuntu 14.04, Titan X Pascal and 12G memory. The experiment first provides the dataset and evaluation protocol, followed by a detailed ablation analysis and an overall evaluation of the proposed method.

### 3.1. Datasets and Evaluation Protocol

DOTA [47] is a large and challenging aerial image dataset in the field of object detection, including 2806 pictures and 15 categories, with a picture scale ranging from $800 \times 800$ to $4000 \times 4000$. It contains a training set, validation set, and test set, which account for 1/2, 1/6, and 1/3 of the total data set, respectively. Among them, the training and validation sets were marked with 188,282 instances, with an arbitrary direction quadrilateral box. In this study, we used the 1.0 version of rotating object detection, and the image was cropped into $600 \times 600$ slices. It was scaled to $800 \times 800$ during the training. Short naming is defined as plane (PL), baseball diamond (BD), bridge (BR), ground field track (GTF), small vehicle (SV), large vehicle (LV), ship (SH), tennis court (TC), basketball court (BC), storage tank (ST), soccer-ball field (SBF), roundabout (RA), harbor (HA), swimming pool (SP), and helicopter (HC). The official evaluation protocol of the DOTA in terms of the mAP is used.

HRSC2016 [48] is a dataset dedicated to ship detection is in the field of object detection. The dataset contains 1061 images from two scenarios, including ships on the sea and ships close to the shore. There are three levels of tasks (for single class, four types, and 19 types of ship detection and identification). The image sizes range from $300 \times 300$ to $1500 \times 900$, and most of them are larger than $1000 \times 600$. Among them, the training set was 436,

the verification set was 181, and the test set was 444. The example is marked by a rotating rectangular box, and the standard evaluation protocol of HRSC2016 in terms of mAP is used.

### 3.2. Implementation Details

The proposed method is implemented based on the rotation detection benchmark proposed by Yang et al [49]. We used RetinaNet as the baseline method and ResNet50, ResNet101, and ResNet152 as the backbone network for the experiments. To reflect the fairness of the experiment, all comparative experiments used the same backbone network, and the batch size was set to 8, owing to the limitation of GPU memory. In all experiments, we use the momentum SGD optimizer to optimize the network, and the momentum and weight decay are 0.9 and $1 \times 10^{-4}$, respectively. The initial learning rate is $5 \times 10^{-4}$, and for each training epoch, the learning rate decays to 0.1 times the original, and the size of the epoch depends on the number of training samples. The hyperparameters $\alpha$ and $\beta$ are set to a sequence (see Figure 8), which gradually decreased as the number of iterations increased.

### 3.3. Ablation Study

The ablation study includes the effect of the modulation mechanism and the constrained mechanism on the network, as well as the convergence and generality of the proposed constrained loss function. (For the convenience of comparison, the validation set in the DOTA is used for evaluation, because the test set label has not been released.)

Effects of the modulation mechanism: We experimented with the proposed decoupling modulation mechanism in the regression loss function, and compared it with popular loss functions, such as $L_1$, smooth-$L_1$, IoU-smooth-$L_1$ [33], and $L_{mr}$ [38]. In the experiment, we used the same backbone network (resnet50) and five-parameter regression method, and used RetinaNet as the baseline method. The experimental results show that in Table 2, our modulation mechanism has achieved better performance than $L_{mr}$ [38], which has increased by 0.4 and 0.6 in DOTA and HRSC2016, respectively. This further proves our idea that decoupling the regression parameters can improve the performance of the network model.

**Table 2.** Ablation experiments of improved modulation mechanism on DOTA and HRSC2016. (Bold indicates the best result in the comparison method).

| Backbone | Loss | mAP (DOTA) | mAP (HRSC2016) |
|----------|------|------------|----------------|
| ResNet-50 | $L_1$ | 61.9 | 80.7 |
| ResNet-50 | smooth-$L_1$ | 62.1 | 81.6 |
| ResNet-50 | IoU-smooth-$L_1$ [33] | 62.7 | 82.7 |
| ResNet-50 | $L_{mr}$ [38] | 64.5 | 83.1 |
| ResNet-50 | $L_{dm}$(ours) | **64.9** | **83.7** |

Effects of the constraint mechanism: To verify the effectiveness of the constraint loss function ($L_C$), we experimented with the center point constraint and the size constraint, respectively and explored the influence of different constraint domains on the performance of the model. In the experiment, the CPs $\alpha$ and $\beta$ are designed to be 5,4,3,2,1,0, and test them, respectively, where 0 indicates an unconstrained state. To ensure fairness of the experiment, each comparison experiment was trained for 30 epochs (training times per epoch was 20,673). In addition, the proposed decoupling modulation loss is used. The experimental results are shown in Figure 9. Obviously, there are some positive and negative effects due to the introduction of $\alpha$ and $\beta$. Compared with the unconstrained case, the effective CPs accelerate the network convergence speed. In particular, when CPs = 4 or 5, the network convergence speed is faster, but the model performance is sacrificed. The network convergence speed and model performance were improved when CPs = 1, 2, or 3. The performance of the model is the best when $\alpha$ = 2 and $\beta$ = 3, which are improved by 3.5 and 3.2, respectively. Although this is not the final result (only 30 epochs are trained),

and this improvement will become smaller as the training expands, this is sufficient to confirm our method. The introduction of CPs can not only improve the performance of the model, but more importantly, it greatly improves the convergence speed of the model and reduces resource consumption.

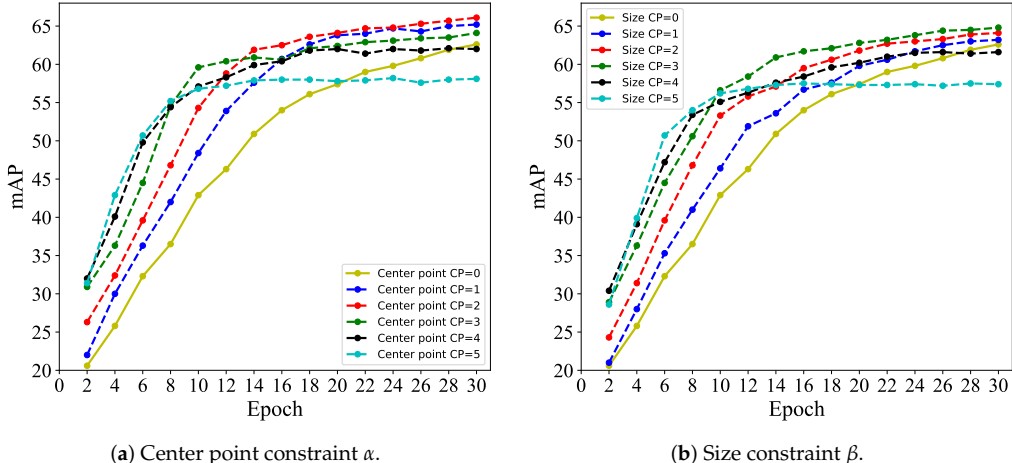

(**a**) Center point constraint $\alpha$.　　　　(**b**) Size constraint $\beta$.

**Figure 9.** Ablation experiments of bounding box center point constraint and size constraint in DOTA. The RetinaNet and ResNet50 as the baseline and backbone, respectively.

Based on the above experiments, we found that appropriate center point constraints and size constraints can have a positive effect on network training. Therefore, it is necessary to explore these combinations. In the experiment, we predefined an optimal combination $(\alpha, \beta) = (2,3)$ (from the ablation experiment of the center point constraint and size constraint), and designed some other combinations. The results are shown in Table 3. The predefined combination $(\alpha, \beta) = (2,3)$ and $L_{dm}$ led to the best performance compared to other combinations and loss functions, which confirmed our idea.

**Table 3.** Combination experiment of center point constraint and size constraint. (Bold indicates the best result in the comparison method).

| Backbone | Loss | $\alpha$ | $\beta$ | mAP (DOTA) | mAP (HRSC2016) |
|----------|------|----------|---------|------------|----------------|
| ResNet50 | $L_{dm}$(ours) | 2 | 3 | **66.9** | **85.2** |
| ResNet50 | $L_{dm}$(ours) | 2 | 2 | 66.4 | 85.0 |
| ResNet50 | $L_{dm}$(ours) | 3 | 3 | 65.9 | 85.0 |
| ResNet50 | $L_{dm}$(ours) | 1 | 4 | 65.2 | 83.7 |
| ResNet50 | $L_{dm}$(ours) | 4 | 1 | 63.6 | 84.4 |
| ResNet101 | smooth-$L_1$ | 2 | 3 | 66.9 | 84.6 |
| ResNet101 | IoU-smooth-$L_1$ | 2 | 3 | 67.6 | 85.2 |
| ResNet101 | $L_{mr}$ | 2 | 3 | 68.4 | 85.6 |
| ResNet101 | $L_{dm}$(ours) | 2 | 3 | **68.7** | **86.1** |

Convergence analysis: We experimented with the network under different CPs and recorded the number of training required for the model to stabilize. The results are shown in Table 4, where different CPs have significant differences in the consumption of training resources of the network. Larger CPs mean greater tolerance for predicting bounding boxes, less training resources are required, and faster convergence and stability, while smaller CPs have the opposite.

**Table 4.** The number of iterations required for the network to reach stability under different parameters.

| Datasets | 5 | 4 | 3 | 2 | 1 | 0 |
|---|---|---|---|---|---|---|
| DOTA | $3.2 \times 10^5$ | $4.2 \times 10^5$ | $6 \times 10^5$ | $6.8 \times 10^5$ | $8.4 \times 10^5$ | $12 \times 10^5$ |
| HRSC2016 | $0.4 \times 10^5$ | $0.42 \times 10^5$ | $0.48 \times 10^5$ | $0.64 \times 10^5$ | $0.8 \times 10^5$ | $1.3 \times 10^5$ |

Adjustability: By combining the characteristics of different CPs, an experiment of cascade constraint sequence was designed. We designed 30 epochs and divided them into six equal parts, defined as A0, ..., A5 = [epoch1, epoch5], ..., [epoch 26, epoch30]. Subsequently, a series of sequence experiments were designed, and the results are listed in Table 5. As the sequence complexity increased, the performance of the model improved. In particular, in the G4 case, the model's mAP was improved by 2.7 compared to the G0 case, which confirms the effectiveness of the cascade constraint sequence.

**Table 5.** Ablation experiment of cascade constraint parameter sequence. The center point constraint $\alpha$ and the size constraint $\beta$ are set to the same value. RetinaNet is the base model, and ResNet50 is the backbone. (Bold indicates the best result in the comparison method).

| Group | A0 | A1 | A2 | A3 | A4 | A5 | mAP (DOTA) | mAP (HRSC2016) |
|---|---|---|---|---|---|---|---|---|
| G0 | 1 | 1 | 1 | 1 | 1 | 1 | 66.1 | 84.2 |
| G1 | 2 | 2 | 2 | 1 | 1 | 1 | 66.3 | 84.9 |
| G2 | 3 | 3 | 2 | 2 | 1 | 1 | 67.4 | 85.5 |
| G3 | 4 | 4 | 3 | 3 | 2 | 1 | 67.6 | 85.7 |
| G4 | 5 | 4 | 3 | 2 | 1 | 0 | **67.9** | **86.2** |

Generalizability: To verify the generality of the constraint loss function, we experimented with our method in the popular regression loss function. In the experiment, RitinaNet was used as the baseline method, and Resnet50 was used as the backbone network. The constraint parameter sequence adopts the G4 sequence in Table 5 owing to its excellent performance. In the experiment, the performance of different regression loss functions in the OBB and HBB tasks of DOTA were tested, and the results are shown in Table 6. Obviously, after the optimization of $L_{dm}$ or CPs proposed by us, the performance of the model has been improved to varying degrees. In particular, it has better performance in the deviation-based method, which increases the Huber loss by 2.4 mAP on the DOTA, and increases the MAE loss by 1.9 mAP on the HRSC2016. One possible explanation is that $L_{dm}$ plays an important role in the OBB task.

Effects of data augmentation: Many studies have proved that data enhancement can effectively improve detector performance. We extended the data by random horizontal, vertical flipping, random graying, random rotation,and random change channels. In addition, additional enhancements have been made to categories with a small number of samples (such as helicopters and bridges). The experimental results are shown in Table 7, and a 3.1% improvement was obtained on the DOTA (from 67.9% to 71.0%); a 1.7% improvement was obtained improvement on the HRSC2016 (from 86.2% to 87.9%). We also explored a larger backbone network, and the results showed that a larger backbone can result in better performance. The final performance of our improvement was 74.3% and 88.9% using ResNet152 as the backbone network. In addition, our training times is $8 \times 10^5$, which is far less than that of the popular method, and the resource consumption due to training as greatly reduced.

**Table 6.** The performance of the constrained loss function on the popular regression loss function. (Bold indicates the best result in the comparison method).

| Task | Loss | $L_{dm}$ | CPs | mAP (DOTA) | mAP (HRSC2016) |
|------|------|----------|-----|------------|----------------|
| OBB | MSE | ✓ | ✓ | 66.2(+1.6) | 82.2(+1.0) |
| OBB | MAE | ✓ | ✓ | 66.3(+1.3) | 82.6**(+1.9)** |
| OBB | Huber [41] | ✓ | ✓ | 67.1**(+2.4)** | 85.2(+1.6) |
| OBB | Log-Cosh [42] | ✓ | ✓ | 65.4(+0.7) | 84.8(+1.2) |
| OBB | Quantile [43] | ✓ | ✓ | 64.7(+0.2) | 84.3(+0.9) |
| HBB | IoU [44] | - | ✓ | 66.5(+1.2) | - |
| HBB | GIoU [45] | - | ✓ | 66.8(+0.6) | - |
| HBB | DIoU [46] | - | ✓ | 67.0(+1.0) | - |
| HBB | CIoU [46] | - | ✓ | 67.4**(+1.8)** | - |

**Table 7.** Ablation experiments of data augmentation on DOTA and HRSC2016. RetinaNet is the base model with the proposed $L_{dm}$ and CPs. (Bold indicates the best result in the comparison method).

| Backbone | $L_{dm}$ + CPs | Data Augmentation | mAP (DOTA) | mAP (HRSC2016) |
|----------|----------------|-------------------|------------|----------------|
| ResNet-50 | ✓ | | 67.9 | 86.2 |
| ResNet-50 | ✓ | ✓ | 71.0 | 87.9 |
| ResNet-101 | ✓ | ✓ | 72.6 | 88.3 |
| ResNet-152 | ✓ | ✓ | **74.3** | **88.9** |

*3.4. Overall Evaluation*

We compare our proposed constraint loss function with the state-of-the-art rotating object detection method on two datasets DOTA [47] and HRSC2016 [48].

Results in DOTA: We first experimented with our method on the DOTA dataset and compared it with popular rotated object detection methods, as depicted in Table 8. The results of the overall evaluation experiment were obtained by submitting our model to the official DOTA evaluation server. In the experiment, the training and verification sets of the DOTA were used as training samples, and the test set was used to verify the performance of the model. The compared methods include scene file detection methods R[2]CNN [50], RRPN [30], popular rotated object detection method ICN [51], RoI Transformer [31], Gliding Vertex [36], and methods that consider sudden changes in loss, such as SCRDet [33], R[3]Det [35], and RSDet [38]. The performance of our method is 1.1 better than the best result in the comparison method (RSDet+ResNet152+Refine). Although our method did not achieve state-of-the-art performance in the DOTA rankings, it showed the best performance in the one-stage detector. In addition, our method saves more than 30% of the computing resources compared with most methods. The visualization results are shown in Figure 10.

**Table 8.** Detection accuracy (AP for each category and overall mAP) on different objects and overall performances with the state-of-the-art methods on DOTA. (Bold indicates the best result in the comparison method).

| Method | Backbone | PL | BD | BR | GTF | SV | LV | SH | TC | BC | ST | SBF | RA | HA | SP | HC | mAP | Times |
|--------|----------|----|----|----|-----|----|----|----|----|----|----|-----|----|----|----|----|-----|-------|
| FR-O [47] | ResNet101 | 79.1 | 69.1 | 17.2 | 63.5 | 34.2 | 37.2 | 36.2 | 89.2 | 69.6 | 59.0 | 49.4 | 52.5 | 46.7 | 44.8 | 46.3 | 52.9 | - |
| R[2]CNN [50] | ResNet101 | 80.9 | 65.7 | 35.3 | 67.4 | 59.9 | 50.9 | 55.8 | 90.7 | 66.9 | 72.4 | 55.1 | 52.2 | 55.1 | 53.4 | 48.2 | 60.7 | - |
| RRPN [30] | ResNet101 | 88.5 | 71.2 | 31.7 | 59.3 | 51.9 | 56.2 | 57.3 | 90.8 | 72.8 | 67.4 | 56.7 | 52.8 | 53.1 | 51.9 | 53.6 | 61.0 | - |
| ICN [51] | ResNet101 | 81.4 | 74.3 | 47.7 | 70.3 | 64.9 | 67.8 | 70.0 | 90.8 | 79.1 | 78.2 | 53.6 | 62.9 | 67.0 | 64.2 | 50.2 | 68.2 | - |
| RoI-Trans [31] | ResNet101 | 88.6 | 78.5 | 43.4 | 75.9 | 68.8 | 73.7 | 83.6 | 90.7 | 77.3 | 81.5 | 58.4 | 53.5 | 62.8 | 58.9 | 47.7 | 69.6 | - |
| SCRDet [33] | ResNet101 | 90.0 | 80.7 | 52.1 | 68.4 | 68.4 | 60.3 | 72.4 | 90.9 | 88.0 | 86.9 | 65.0 | 66.7 | 66.3 | 68.2 | 65.2 | 72.6 | $1.35 \times 10^6$ |
| Gliding Ver [36] | ResNet101 | 89.9 | 85.9 | 46.1 | 78.5 | 70.3 | 69.4 | 76.9 | 90.7 | 79.3 | 83.8 | 57.8 | **68.3** | **72.9** | **71.0** | 59.8 | 73.4 | $1.27 \times 10^6$ |
| R[3]Det [35] | ResNet152 | 89.2 | 80.8 | 51.1 | 65.6 | **70.6** | 76.0 | 78.3 | 90.8 | 84.9 | 84.4 | **65.1** | 57.2 | 68.1 | 68.9 | 60.9 | 72.8 | $1.35 \times 10^6$ |
| RSDet [38] | ResNet152 | 90.1 | 82.0 | 53.8 | 68.5 | 70.2 | 78.7 | 73.6 | 91.2 | 87.1 | 84.7 | 64.3 | 68.2 | 66.1 | 69.3 | 63.7 | 74.1 | $1.08 \times 10^6$ |
| Ours | ResNet50 | 89.5 | 86.6 | 49.6 | 64.8 | 57.1 | 65.1 | 68.5 | 90.1 | 82.9 | 86.6 | 62.8 | 65.0 | 62.5 | 68.8 | 58.4 | 71.0 | $7.2 \times 10^5$ |
| Ours | ResNet101 | 90.1 | **87.7** | **55.6** | 72.6 | 63.5 | 72.8 | 72.2 | 90.1 | 87.7 | 88.0 | 63.5 | 65.3 | 64.1 | 67.8 | 65.8 | 73.7 | $8.1 \times 10^5$ |
| Ours | ResNet152 | **90.2** | 87.5 | 53.8 | 73.1 | 70.6 | **79.4** | 77.3 | **91.0** | **88.7** | 87.5 | 65.0 | 67.7 | 68.8 | 69.7 | **68.9** | **75.2** | $8.1 \times 10^5$ |

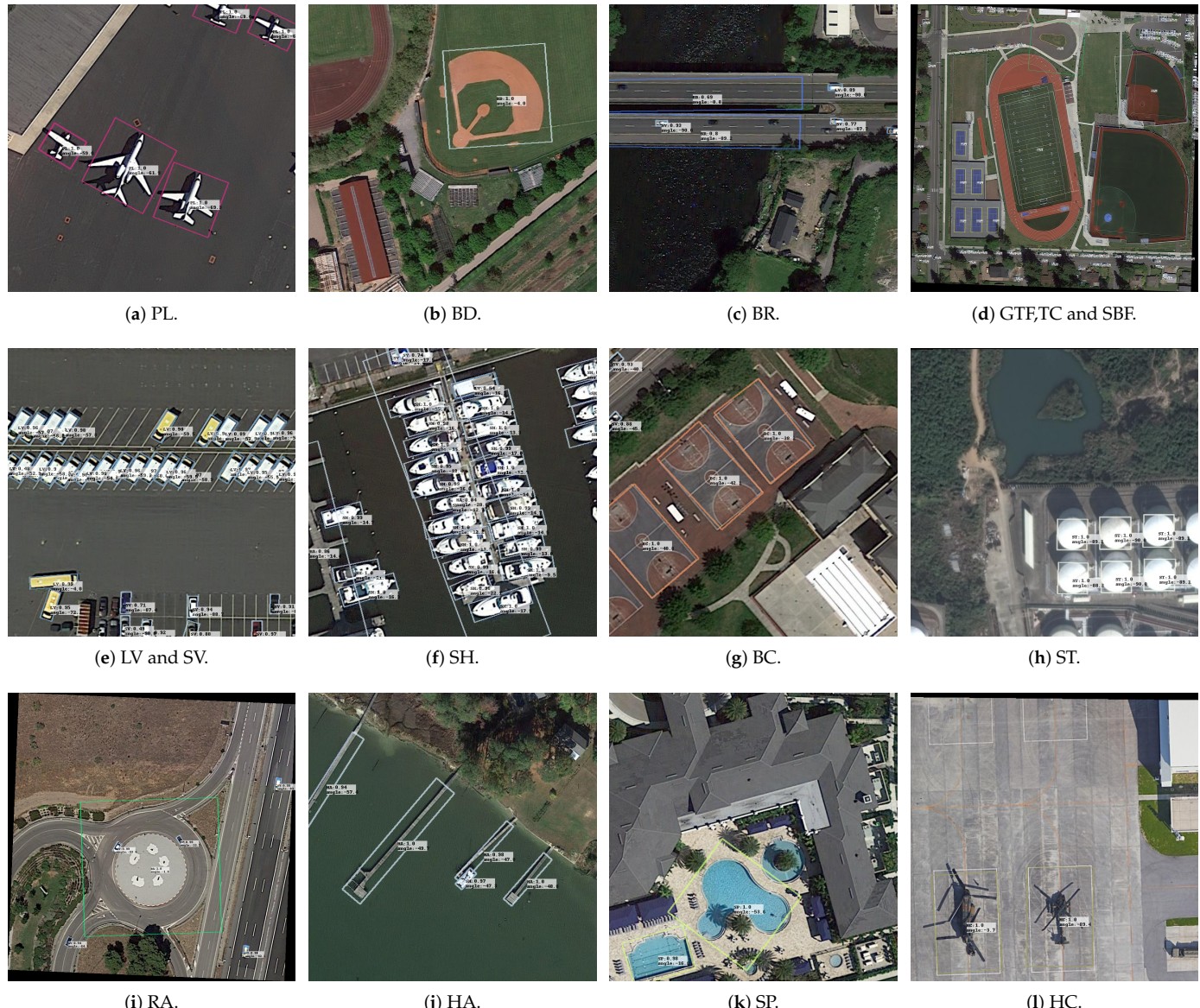

**Figure 10.** Some detection results of the our method on DOTA. The arbitrary-oriented objects are correctly detected. The category short naming is defined as: Plane (PL), Baseball diamond (BD), Bridge (BR), Ground field track (GTF), Small vehicle (SV), Large vehicle (LV), Ship (SH), Tennis court (TC), Basketball court (BC), Storage tank (ST), Soccer-ball field (SBF), Roundabout (RA), Harbor (HA), Swimming pool (SP) and Helicopter (HC).

Results in HRSC2016: We also experimented with our method in HRSC2016 and compared it with popular detectors, and the results are shown in Table 9. First, a comparative experiment was carried out using the methods proposed in scene text detection, RRPN, and R2CNN, and the detection accuracy was not ideal. RoI Transformer and Gliding Vertex have achieved good detection accuracy but require more training resources. RetinNet-H and RetinaNet-R were used as baseline methods, among which RetinaNet-R obtained 89.1% detection accuracy. R³Det [35] achieved a better detection accuracy than the above method. In the end, our method achieved an accuracy of 89.7%, achieved the state-of-the-art performance under the optimization of $L_{dm}$ and CPs, and nearly half of the training resources were saved. The visualization results are shown in Figure 11.

**Table 9.** Detection accuracy and the number of iterations on HRSC2016. Times represents the number of trainings required for the model to achieve the best performance. (Bold indicates the best result in the comparison method).

| Method | Backbone | Loss | Size | mAP | Times |
|---|---|---|---|---|---|
| R2CNN [50] | ResNet101 | - | $800 \times 800$ | 73.1 | - |
| RRPN [30] | ResNet101 | - | $800 \times 800$ | 79.1 | - |
| R$^2$PN [52] | VGG16 | - | - | 79.6 | - |
| RetinaNet-H [9] | ResNet101 | smooth-$L_1$ | $800 \times 800$ | 82.9 | $2.4 \times 10^5$ |
| RRD [53] | VGG16 | - | $384 \times 384$ | 84.3 | - |
| RoI-Transformer [31] | ResNet101 | $L_1$ | $512 \times 800$ | 86.2 | $3.0 \times 10^5$ |
| Gliding Vertex [36] | ResNet101 | smooth-$L_1$ | - | 88.2 | $3.6 \times 10^5$ |
| RetinaNet-R | ResNet101 | smooth-$L_1$ | $800 \times 800$ | 89.1 | $3.6 \times 10^5$ |
| R$^3$Det [35] | ResNet101 | SkewIoU | $800 \times 800$ | 89.2 | $2.6 \times 10^5$ |
| RSDet [38] | ResNet101 | $L_{mr}$ | - | 86.5 | $2.6 \times 10^5$ |
| Ours | ResNet101 | $L_{dm}$ | $600 \times 600$ | 88.7 | $2.4 \times 10^5$ |
| Ours | ResNet101 | $L_{dm}$ | $800 \times 800$ | 89.1 | $2.4 \times 10^5$ |
| Ours | ResNet101 | $L_{dm}$ + CPs | $800 \times 800$ | 89.5 | $1.4 \times 10^5$ |
| Ours | ResNet152 | $L_{dm}$ + CPs | $800 \times 800$ | **89.7** | $\mathbf{1.4 \times 10^5}$ |

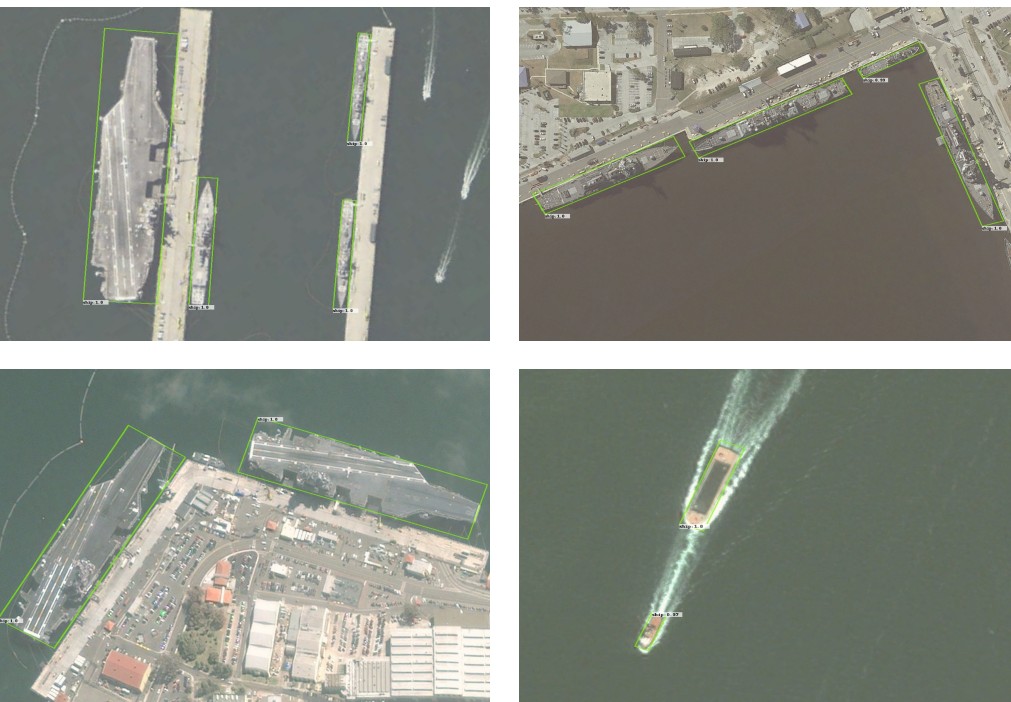

**Figure 11.** Ship detection results on the HRSC2016 benchmarks. The green bounding box indicate the prediction box.

## 4. Conclusions

In this study, a constraint loss function is proposed, including the decoupling modulation loss and cascade constraint mechanism. The former overcomes the regression to the rotating frame during the loss mutation process, and the latter enables the direction of a parameter update to be supervised and improves the convergence speed of the network. As a result, our method enables the method based on RetinaNet-R to achieve a 75.2% performance on the DOTA benchmark dataset, and achieves a state-of-the-art performance on the HRSC2016 dataset. In addition, the cascade constraint mechanism is applied to the popular regression loss function to achieve a better performance.

In this study, we improved the performance of the regression loss function through the modulation mechanism and the constraint mechanism, which not only improved the performance of the model, but also saved training resources. However, there are also the following limitations:

- The selection method of constraint parameters is supervised and artificially set, which limits its performance.
- Although the cascade constraint sequence is variable, the range of each stage is fixed (as shown in Table 5 A0 = [epoch1∼epoch5]). Ideally, the range of each stage should be learnable.
- This article adds the constraint domain as the center point and size of the bounding box, but the rotation angle is ignored. Better results can probably be obtained if a constraint domain similar to the Gaussian radius is added to the rotation angle.

However, it is worth noting that the selection method of constraint parameters in the proposed method is supervised and artificially set, which limits its performance.

Therefore, in future research, we intend to try a learnable constraint parameter selection mechanism to improve the generalization ability of the model. In addition, a future research direction is to explore a new way of defining a rotated bounding box. The existing methods are all rotating rectangular boxes or four vertices. The former has the problem of angular periodicity, and the latter is easily affected by the order of the vertices. Therefore, the study of a better and effective rotating bounding box definition method is of great significance to the development of rotated object detection.

**Author Contributions:** Conceptualization, L.Z. and L.W.; methodology, L.Z.; validation, H.W., L.W.; formal analysis, L.W.; investigation, L.Z.; resources, H.W. and C.P.; data curation, Q.L. and C.P.; writing—original draft preparation, L.Z.; writing—review and editing, L.Z. and X.W.; visualization, L.Z. and X.W.; supervision, H.W.; project administration, H.W.; funding acquisition, L.W. All authors have read and agreed to the published version of the manuscript.

**Funding:** This work was supported by the National Natural Science Foundation of China under Grants (61773377), the Nondestructive Detection and Monitoring Technology for High Speed Transportation Facilities, Key Laboratory of Ministry of Industry and Information Technology, and the Fundamental Research Funds for the Central Universities, (NO.NJ2020014).

**Institutional Review Board Statement:** Not applicable.

**Informed Consent Statement:** Not applicable.

**Data Availability Statement:** The data presented in this study are available on request from the first author.

**Conflicts of Interest:** The authors declare no conflict of interest.

## Appendix A. Details of Gradient Analysis

For the input image $X \in [W \times H \times 3]$, after network processing, output $Y_i \in [x, y, w, h, \theta]$, where $i = 1, 2, 3 \ldots n$, represents the corresponding proposal. Assuming that the ground truth of the prediction box $Y^* = [x^*, y^*, w^*, h^*, \theta^*]$, we obtain the deviation of the center point, size, and angle of the target frame ($L_1, L_2, L_3$) according to our design (2)–(4). We assume that $W$ represents the set of all weight parameters, $b$ is the deviation, and n is the number of samples. Forward propagation process $z = WX + b$, activation function $y = \sigma(z)$.

(1) The first is the calculation of the partial derivative of the loss function $S(L)$ with respect to the output layer.

$$S(L) = \begin{cases} 0.5L^2 & if\, |L| < 1 \\ |L| - 0.5 & otherwise \end{cases} \tag{A1}$$

$$\frac{\partial S}{\partial L} = \begin{cases} L & if |L| < 1 \\ \pm 1 & otherwise \end{cases} \tag{A2}$$

From the chain rule:

$$\delta_j^Y = \frac{\partial S}{\partial z_j^Y} = \frac{\partial S}{\partial L_j^Y} \frac{\partial L_j^Y}{\partial z_j^Y} = L \odot \sigma'(z^Y) \tag{A3}$$

$$\delta^Y = \bigtriangledown L \odot \sigma'(z^Y) \tag{A4}$$

where $\bigtriangledown$ represents the gradient vector of the loss function S with respect to the predicted value Y, $\sigma'$ represents the partial derivative of the activation function $z(X)$, and $\odot$ is the *Hadamard* product, which represents the point-to-point multiplication operation between matrices or vectors.

(2) To calculate the partial derivative of the FC, according to the above analysis, the partial derivative of the *j*-th element of the *l*-th layer can be expressed as follows:

$$\begin{aligned} \delta_j^l &= \frac{\partial S}{\partial_j^l} = \sum_k \frac{\partial S}{\partial z_k^{l+1}} \frac{\partial z_k^{l+1}}{\partial_j^l} = \sum_k \frac{\partial z_k^{l+1}}{\partial_j^l} \delta_k^{l+1} \\ &= \sum_k W_{kj}^{l+1} \delta_k^{l+1} \sigma'(z_j^l) \end{aligned} \tag{A5}$$

The vector form is expressed as:

$$\delta^l = ((W^{l+1})^T \delta^{l+1}) \odot \sigma'(z^l) \tag{A6}$$

During the parameter update process, the partial derivative of parameters W and b can be expressed as

$$\frac{\partial S}{\partial W_{jk}^l} = Y_k^{l-1} \delta_j^l \tag{A7}$$

$$\frac{\partial S}{\partial b_j^l} = \delta_j^l \tag{A8}$$

(3) The pooling layer compresses the input during the forward propagation process. The input $P^{l-1}$ of the pooling layer can be obtained from $P^l$. This process is usually called an upsample.

$$P^{l-1} = upsample(P^l \odot \sigma'(z^{l-1})) \tag{A9}$$

where the first term represents upsampling, and the second term is the derivative of the activation function. The second term can be understood as the constant 1 in the pooling process, because no activation function is involved in the pooling layer. In addition, there is no parameter update in the pooling layer because the W and b parameters are not involved.

(4) Similar to the back propagation process of the pooling layer, the input $C^{l-1}$ of the convolutional layer can be obtained from $C^l$.

$$C^{l-1} = C^l \frac{\partial z^l}{\partial z^{l-1}} = C^l * rot180(W^l) \odot \sigma'(z^{l-1})) \tag{A10}$$

During the parameter update process, the gradient of W and b can be expressed as:

$$\frac{\partial J(W,b)}{\partial W^l} = \frac{\partial J(W,b)}{\partial z^l} \frac{\partial z^l}{\partial W^l} = a^{l-1} * C^l \tag{A11}$$

$$\frac{\partial J(W,b)}{\partial b^l} = \sum_{u,v} (C^l)_{u,v} \tag{A12}$$

Based on the above analysis, we found that the partial derivative of the loss function $S$ with respect to $L$ is always equal to the prediction deviation $L$ (see (A2)) and is linearly positively correlated with the gradient of the output layer elements (see (A4)). This means that the introduction of the constraint loss function will have a direct impact on the back propagation process.

When the center point constraint ($L_1^*$) or the size constraint ($L_2^*$) is activated, the components of the regression deviation $L_C$ are reduced, and this response is directly transferred to the calculation of the gradient value of the main layer ($\delta^Y$, $\delta^l$, $P^{l-1}$, and $C^{l-1}$), and parameter adjustment without additional steps. This simplifies the backpropagation task. In particular, it is more pronounced when $L_1^*$ and $L_2^*$ are activated simultaneously. This means that only the angle parameter $\theta$ is adjusted ($L = L_3$), and the task will be simple and easy to implement. It is worth noting that when $L_1^*$ and $L_2^*$ are activated, the center point and size of the candidate frame are adjusted to the constraint range (see Figure 6), and subsequent adjustments have also been suppressed. This avoids additional calculations and development in unfavorable directions.

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
