# Peer review of "Constraint Loss for Rotated Object Detection in Remote Sensing Images"

_remotesensing, doi:10.3390/rs13214291_

Round 1
Reviewer 1 Report
I am glad to see the authors' improvements - I am mostly happy with the submission now. Small changes I would suggest:
- "Proof of Gradient Analysis" makes no sense. One cannot prove analysis. Rephrase.
- Broken link in line 368 "Eq.??"
- Do not write "Eq. (1)" but just "(1)" which is in line with the style norms.
Reviewer 2 Report
Dear Authors,
Many thanks for your updated manuscript submission to MDPI Journal of Remote Sensing. Comparing this version and the prior version, I think the authors has significantly improved the overall quality of this research article. Adding specific details, I evaluated this paper in the following aspects, while a few issues requires further improvements (not limited to) as below:
a) This paper establiashed a constraint loss function (which includes the decoupling modulation loss and cascade constraint mechanism) to handle the research task on rotational object detection for remote sensing images. The authors show that their approach achieved 75.2% mAP in DOTA dataset and state-of-the-art results in HRSC2016 dataset. They also specified that this method improved the performance of regression loss function while the limitations are due to supervised classification and artificial set-up on parameters. The overall organization of this paper is quite good till now.
b) Regarding the research topic on oriented object detection, it is both special and technical ideas are refresh. However, I think the authors may have neglected to cite and include the following research articles:
X.-X. Zhang, and X. Zhu, “Moving vehicle detection in aerial infrared image sequences via fast image registration and improved YOLOv3 network,” Int. J. Remote Sens., vol. 41, no. 11, pp. 4312-4335, 2020.
W.-T. Liao, X. Chen, J.-F. Yang, S. Roth, M. Goesele, M. Y. Yang, and B. Rosenhahn, “LR-CNN: local aware region CNN for vehicle detection in aerial imagery,” ISPRS Annals. Photogram., Remote Sens. Spatial Inform. Sci., vol. V-2-2020, 2020., pp. 381-388, 2020.
J.-F. Lei, Y.-X. Dong, and H.-G. Sui, “Tiny moving vehicle detection in satellite video with constraints of multiple prior information,” Int. J. Remote Sens., vol. 42, no. 11, pp. 4110-4125, 2021.
Meanwhile, by inspecting the References on list, I think the authors should include more latestly published articles, i.e., Trans. Image Processing, Trans' Geoscience & Remote Sensing and Trans'PAMI (i.e., concealed object detection) in Years 2019-2021 to enhance theoretical background on review.
c) In general, the authors have improved the methodology and technical presentations of their work, their approach is updated with contributions to the area of object detection for low-resolution aerial images in the area of Remote Sensing. I guess the following issues still need to be updated:
i) Adding more specific details on the statements of major contributions in the Introduction Section;
ii) Can you explain that why the number of iteration times is x 10^5 level?
iii) The metrics on some tables are not clearly stated or labeled (i.e., AP or mAP in Tables 3, 5, 6 and 7?)
iv) Minor typos in Tables 8-9 (top-left): Mothed --> Method.
v) I don't think the texts in the text frames of Figure 7 should be italic; only the italic notations of unknown parameters (last box) should be retained.
d) The conclusions are consistent with the evidence and arguments presented, which addressed the main research tasks as posed. However, I think this paper either missed a short subsection discussion the limitations of their study (only a short sentence on statement), or specify their parallel comparsion to some other related work (as I listed above) in the 2nd paragraph of conclusion section to fill up any possible shortcomings.
e) ResNet-XXX are not the only available models to conduct experiments and Ablation study; the current section looks just OK.
f) Regarding References, in addition to the problematic issues as specified above, the current version is absent from abbreviated formats; also, Refs. [1] and [45] are different from the uniformly cited styles, please fix both of them; another issue is the chaos on capital / uncapital letters on some journal citations, which must be corrected in the proof version.
g) Besides, I think while the use of English is overall acceptable, there are quite some grammatical mistakes (and minor typos) remaining in this version, i.e., Result -> Results, Effect --> Effects, Mothed --> Method, etc. Some sentences of transit paragraphs are also rough and hard in phrase connections. Please invite a native English speaker to fix the literal issues.
Thanks again for your hard work. We look forward to reviewing your updated manuscript into final acceptance. Good luck and stay well!
Best regards,
Yours faithfully,
Author Response
Please see the attachment

This manuscript is a resubmission of an earlier submission. The following is a list of the peer review reports and author responses from that submission.
Round 1
Reviewer 1 Report
Dear Authors,
Many thanks for your manuscript submission to Journal of Remote Sensing. After careful review, I think this paper requires at least one round of major revision, despite of the specified contributions and some other merits therein. The possible major and minor issues can be summarized as below:
Major problems of this manuscript:
a) The authors proposed a contraint loss function to overcome the related regression issues, and claimed the process of loss mutation may supervise the parameter updates and speed up the network performance. I think the authors need to be more specific on the originality of this constraint loss function, and how it contributes to improve the performance in different manners. You may need to compare some parallel work in Years 2018-2021.
b) The backbones in use are mainly ResNet101, a few are using ResNet50, ResNet152 or VGG16. I think some parallel work claimed that ResNet50 is good enough, while not many significant improvements on mAP as displayed in your approach (seen from Table 8), please consider explaining it.
c) The two datasets are DOTA and HRSC2016. Imbalanced experiments were done for the two datasets, in other words, I think the authors need to supplement more work on HRSC2016 to convince the reviewers for the validity and efficiency of your approach, then specify that there is NO cross-dataset difference among your experimental design.
d) I think that it had better to align the ablation experiments into the section of experiments in your Section 4 (Results and Discussions).
e) The Introduction Section and Conclusion Section are a bit too generic. I think the Intro. part should be requiring a major rewrite, and the conclusion section should include main summary of your work, limitations of study, opening questions, and prospective study for future research orientations. Hence, consider expanding the paragraphs 1-2 in Section 1 with more specific details, and split the single paragrah of Section 5 into 3 paragraphs (main summary, discussions, and future work), which could be much better.
Minor issues of this manuscript:
a) In the context, all the cited references, figures, tables and equations are labeled with "[?]" or "?", it might result from the format conversion issues, please carefully fix each of these problems in your revised version.
b) Please apply middle alignment for each of the tables in your context.
c) Please include some specific quantitative results in the Abstract session, and remove unnecessary statements distracted from your method.
d) References: each of the cited papers should be numbered, and need to comply with Remote Sensing template format (either abbreviated terms and long-style for conferences); meanwhile, the authors may consider applying some more latest work in Years 2019-2021, typically in Year 2021.
e) Some of the subsections on this manuscript were poorly written; please improve the use of English, calibrate any of the grammatical issues so as to enhance the comprehensive literal quality of this research article.
We look forward to reviewing your updated manuscript after major revision. Stay well and best of luck!
With warm regards,
Yours faithfully,
Reviewer 2 Report
This manuscript addresses rotation invariant object detection for remote sensing images. The modifeid algorithms from [26] are proposed.
The experimental results with two datasets show a slight improvement in accuracy, but a significant improvement in computation time.
[1] First of all, more datasets should be tested.
In addition, the mansucript is confusing in organization and terminologies as follows
[2] Abstract: L1 is differrent from Smooth L1. Authors should use one of them throught out the paper.
[3] Abstract: The open source library (OpenCV) is very extensive. Authors should be more specific about OpenCV's algorithm.
[4] The conditional function should be defined in line 111.
Reviewer 3 Report
Overall, I am rather happy with the substance of this submission. The problem addressed by the authors is relevant, the specific issues identified are correctly motivated and convincing, and the authors' approach is sound and sufficiently novel. The results are also suitably convincing.
Sadly, despite my high ratings of the substance above, from the very first word (literally!) of the manuscript, one cannot help but notice the elephant in the room: the low quality of English. Normally, I would permit this to go onto the next reviewing stage and offer the authors the chance to correct errors, but in this case there is just too much change needed and the current state of affairs makes reading really confusing - far too much for any reputable journal to risk publishing it.
Like I said, confusing errors start from the very first word, namely "rotating". I found this rather difficult to understand until at some point in the manuscript I inferred that the authors mean "rotated" - a very different thing in this context, which changes substantially the topic of the conversation. Nevertheless, even that word is unsuitable and unclear. Rotated relative to what? That is never explained properly.
There are more really confusing things in this submission. For example, the authors' abrupt introduction of openCV. The authors motivate a conceptual problem and then all of a sudden toss openCV on the table, without any explanation. What if I do not use openCV? Is the problem still relevant? This is very poorly executed.
Errors in grammar and more widely in English are pervasive - there are just so many that I will give just an example or two:
- "The deviation-based method is to calculate..." - poor linguistically in several ways.
- "...the ground true" - "true" should be "truth"
- "However, the horizontal bounding box cannot provide accurate orientation and scale information in remote sensing image object detection tasks" - erroneous article use.